# Structure and Function of Dynein’s Non-Catalytic Subunits

**DOI:** 10.3390/cells13040330

**Published:** 2024-02-11

**Authors:** Lu Rao, Arne Gennerich

**Affiliations:** Department of Biochemistry and Gruss Lipper Biophotonics Center, Albert Einstein College of Medicine, Bronx, NY 10461, USA

**Keywords:** cytoplasmic dynein-1, cytoplasmic dynein-2, axonemal dynein, intermediate chain, light intermediate chain, light chain, molecular motors, microtubules

## Abstract

Dynein, an ancient microtubule-based motor protein, performs diverse cellular functions in nearly all eukaryotic cells, with the exception of land plants. It has evolved into three subfamilies—cytoplasmic dynein-1, cytoplasmic dynein-2, and axonemal dyneins—each differentiated by their cellular functions. These megadalton complexes consist of multiple subunits, with the heavy chain being the largest subunit that generates motion and force along microtubules by converting the chemical energy of ATP hydrolysis into mechanical work. Beyond this catalytic core, the functionality of dynein is significantly enhanced by numerous non-catalytic subunits. These subunits are integral to the complex, contributing to its stability, regulating its enzymatic activities, targeting it to specific cellular locations, and mediating its interactions with other cofactors. The diversity of non-catalytic subunits expands dynein’s cellular roles, enabling it to perform critical tasks despite the conservation of its heavy chains. In this review, we discuss recent findings and insights regarding these non-catalytic subunits.

## 1. Introduction

Dynein, a microtubule-based molecular motor, moves towards the minus ends of microtubules (MTs) [1]. This ancient protein complex [2] has undergone evolutionarily diversification into three distinct subfamilies, categorized by their cellular locations: cytoplasmic dynein-1, which carries out various functions in the cytoplasm; cytoplasmic dynein-2, responsible for the retrograde movement of intraflagellar transport (IFT) particles in cilia; and axonemal dyneins, which form the inner and outer rows of arms associated with the doublet microtubules of motile cilia that drive flagellar beating [1,3]. Despite this functional divergence, all dimerized dyneins exhibit a conserved structural organization, comprising two heavy chains and multiple associated smaller subunits. Notably, axonemal dyneins exhibit greater divergence with 1–3 heavy chains, which are discussed in Section 4, “Axonemal Dyneins”.

The heavy chain (HC), with a molecular weight of approximately 500 kDa, is the largest and catalytic subunit of the dynein complex and belongs to the protein family known as ATPases associated with diverse cellular activities (AAA+) [4,5]. The HC’s N-terminus is characterized by a tail region composed of multiple helical bundles (HBs), which provide a scaffold for the attachment of additional subunits [6,7,8,9,10]. Conversely, the C-terminal domain forms a ring-shaped motor domain (or “head”) that is responsible for adenosine triphosphate (ATP) hydrolysis, a critical process that transforms chemical energy into mechanical motion and force. The motor domain is organized into six AAA+ domains that form a ring (AAA1–AAA6) [11,12,13]. Domains AAA1 through AAA4 are capable of nucleotide binding, whereas AAA5 and AAA6 have lost this function [12,13,14,15,16,17,18]. AAA1 serves as dynein’s primary ATPase, and its activity has been demonstrated to be regulated by AAA3 and AAA4 in cytoplasmic dynein-1 [19,20,21,22,23,24]. AAA2, despite its ability to bind ATP, does not exhibit enzymatic activity, suggesting that its role may be to facilitate rigid body motions within the ring [5,25]. While details regarding the allosteric regulation of AAA domains in cytoplasmic dynein-2 and axonemal dyneins are less understood, it is likely that their regulation is similar to that of cytoplasmic dynein-1 due to the conservation of the motor domain in dynein heavy chains [26,27]. Nevertheless, further studies are necessary to reveal their distinct characteristics.

Beyond the ring structure, the motor domain of dynein features several distinct structural elements. A “linker” domain, originating from AAA1, undergoes conformational changes that include docking and undocking from the ring domain, depending on the nucleotide states of dynein’s active ATPase domains [9,11,12,13,28,29]. This dynamic process induces a rotation of the ring [11,28,30], leading to dynein’s directed motion towards the microtubule minus-end [31]. Protruding from AAA4 is a coiled coil, approximately 13 nm long, known as the “stalk” [32,33]. This stalk terminates in a globular microtubule-binding domain (MTBD) that directly binds to MTs [34,35,36,37]. The sliding movement of the stalk’s anti-parallel coiled coils plays a key role in regulating the MTBD’s affinity for MTs [9,18,38,39,40,41,42]. Additionally, a shorter coiled coil, known as the “strut” or “buttress”, extends from AAA5 and interacts with the stalk [12,13,18]. At least for cytoplasmic dyneins, this interaction is crucial for relaying conformational changes within the motor domain to the stalk, thereby modulating the MTBD’s binding affinity for the MTs [12,13,18,41]. For example, the docking and undocking of the linker control the conformational changes in the buttress, thereby influencing the MT-affinity changes in the MTBD in cytoplasmic dynein-1 [41].

While the HC—particularly its motor domain in cytoplasmic dyneins—has been thoroughly examined through structural, biochemical, and single-molecule studies, the non-catalytic subunits of dynein, including intermediate chains (ICs), light intermediate chains (LICs), and light chains (LCs) (Table 1), have received less attention. Until recently, there was a lack of understanding of the molecular mechanisms by which these non-catalytic subunits contribute to dynein’s functionality, both in vitro and in vivo. However, with the advances in cryo-electron microscopy (cryo-EM) techniques, the architectures of full dynein complexes have been elucidated in detail. These findings shed light on the structural and functional roles played by these subunits within dynein complexes. In this review, we delve into recent studies and findings related to these subunits.

## 2. Cytoplasmic Dynein-1

Cytoplasmic dynein-1 (hereafter referred to as dynein-1) is the primary MT-minus-end-directed motor protein that performs a diverse array of critical functions in the cytoplasm of animal cells [1]. Its roles include endocytic trafficking [46], axonal retrograde transport [47,48], mitotic spindle positioning [49,50,51], intricate kinetochore functions [52], and nuclear migration [53,54]. Dynein-1 also plays a role in the intracellular transport of viruses [55], such as adenovirus [56] and human immunodeficiency virus [57].

As a 1.4 MDa complex, dynein-1 consists of a homodimer of HCs, ICs, LICs, and various numbers of LCs depending on the species. The dynein-1 complex of *Saccharomyces cerevisiae* is active and processive in vitro by itself [58], whereas mammalian dynein-1 exhibits only diffusive behavior on MTs [59,60,61]. It was later discovered that mammalian dynein-1 assumes an autoinhibited conformation when isolated (Figure 1a) [6,62]. This autoinhibition is relieved, resulting in a highly processive state, by the addition of the megadalton cofactor dynactin and a coiled-coil cargo adaptor (Figure 1d) [60,61]. Interestingly, even though yeast dynein is active on its own, it harbors a similar autoinhibitory mechanism to that of mammalian dynein-1 [63], suggesting that this regulatory feature may be universal among dynein-1 motor proteins. A wide myriad of dynein-1 adaptors have been identified and studied [64,65,66], highlighting the modular nature of the dynein transport machinery. This modularity allows cells to utilize the same molecular machine to transport various cargoes, demonstrating the adaptability and versatility of dynein-1 in carrying out a broad spectrum of cellular functions.

### 2.1. The Structural Architecture of Dynein-1

The assembly of the dynein-1 complex revolves around the HC, which is the largest subunit within dynein-1 (Figure 1a,d). The C-terminal motor domain of the HC is responsible for enzymatic activity, while its N-terminal tail, comprising a series of relatively flexible HBs, acts as a platform for the attachment of other non-catalytic subunits. Both IC and LIC make direct contact with the HC: the IC binds at the HC’s HB 4 and 5 [7], and the LIC binds at HB 6, positioning itself between the IC-binding site and the motor domain [7,68]. Mammalian dynein-1 features three different LCs that associate with the ICs, facilitating their dimerization. In the autoinhibited state, the IC-LC complex wraps around the HC, with the LCs making contact with the HC’s neck [6]. This interaction may contribute to the autoinhibition of dynein-1.

In the presence of dynactin and an adaptor, dynein is assembled into a tripartite dynein–dynactin–adaptor (DDA) complex (Figure 1d). Structural studies have shown that the coiled-coil adaptor aligns along the short actin filament of dynactin, with dynein interacting with both dynactin and the adaptor [7,67,68,69,70]. Importantly, the stoichiometry of the dynein–dynactin–adaptor complex is not a strict 1:1:1 ratio. A single dynactin can recruit two dyneins and two adaptors when the complex binds to MTs [7,67,70]. Once the complex forms, the two motor domains of the HCs adopt a parallel orientation (Figure 1d), which contrasts with the antiparallel arrangement and crossing stalks seen in the autoinhibited conformation (Figure 1a). As a result, both the stalk and the MTBD are parallel and ready to engage with MTs. In this active conformation, the IC-LC subcomplex of dynein-1 trails behind HCs, unlike in the autoinhibited conformation, where it wraps around the HCs [68].

### 2.2. Dynein-1 Intermediate Chain (IC)

#### 2.2.1. Structure of IC

The IC of dynein-1 is a protein of approximately 74 kDa, characterized by its C-terminal region, which contains tryptophan–aspartic acid (WD) 40 repeats—hereafter referred to as WD repeats. These repeats form a β-propeller ring structure (Figure 2a) that anchors to the HC. The N-terminus of the IC features a single α-helix, followed by an intrinsically disordered region (IDR), which serves as a recruitment site for all the LCs.

The WD repeat is one of the most common protein folds [71,72]. It serves essential roles in many critical biological functions, offering multiple large surfaces for protein–protein interactions [71,72]. In the case of IC, the WD domain rigidly docks to the HC tail, creating a stable platform for LCs and other binding partners. This interaction is also thought to prevent the HCs from aggregating [59], likely because the WD repeats shield the hydrophobic sites on the HC tail. Although the WD repeats are generally considered a scaffold, the possibility of additional, as-yet-undiscovered functions within the IC remains open to future research.

The N-terminus of the IC forms a single α-helix (SAH), followed by a second, shorter nascent helix (H2) [73], both of which are also predicted by AlphaFold [74]. This region can interact with the N-terminal coiled-coil region of dynein-1′s cofactor Nde1/Ndel1 and with a coiled-coil region (CC1B) of dynactin’s largest subunit, p150^Glued^ [75,76,77,78,79]. Despite both Nde1 and p150^Glued^ sharing the same binding region on the IC, they exhibit distinct binding modes [78,80]. Nde1 interacts exclusively with SAH, resulting in the induction of disorder within H2 [81], whereas p150^Glued^ interacts with both SAH and H2, which stabilizes the helical structure of H2 [80]. Predictions based on ColabFold [82] indicate that while the nascent H2 is oriented away from the Nde1 coiled-coil (Figure 3a), it directly engages with the CC1B of p150^Glued^ (Figure 3b). This differential binding confers a higher affinity of the IC for p150^Glued^ as compared to Nde1, with implications for the IC’s function (discussed below). Additionally, the SAH has the potential to fold back on itself to form a third helix within IC, thus inhibiting its interaction with Nde1 and p150^Glued^ in the absence of LCs [80]. This suggests an intrinsic regulatory mechanism that prevents the IC from interacting with Nde1/Ndel1 or p150^Glued^ unless it is part of dynein complexes through the assistance of LCs.

Following the helices, IC contains a stretch of IDR to which the dimeric LCs bind [83,84]. In human dynein, three different dimeric LCs bind to IC in a 1:1 ratio. These are Tctex1, LC8, and Roadblock-1, arranged in the order of their binding sites on the IC from the N-terminus to the C-terminus (Figure 2a). In contrast, for dynein-1 in *Saccharomyces cerevisiae*, only one type of LC (LC8) is involved, with two LC8 dimers binding to the IC [85,86]. This illustrates the species-specific variation in IC-LC interactions. When LCs bind, the IDR transitions towards a more structured conformation [87]. A crystal structure of LCs with IC-derived peptides shows that the IC peptides are positioned on the outside of Tctex1 and LC8, flanking the central β-sheet [86,88,89]. Roadblock-1, on the other hand, interacts with two amphipathic helices of the IC that lie adjacent to the WD repeats [90]. This interaction has also been revealed in the cryo-EM structure of the dynein-1 complexes [6,7,67].

#### 2.2.2. Function of IC

The interaction between the IC’s SAH and Ndel1 is crucial for Ndel1/Lis1-mediated dynein activation [91]. Lis1, a dynein activator, enhances the assembly of DDA complexes [92,93]. Impairment of Lis1’s function leads to lissencephaly, making the understanding of how Lis1 regulates dynein a subject of significant interest (reviewed in [94]). Nde1/Ndel1 has been shown to enhance Lis1’s binding to dynein but inhibits dynein’s movement at high concentrations [95,96]. The N-terminal coiled-coil of Nde1/Ndel1 binds to the IC’s SAH [77,91], while both AlphaFold-based modeling [91,96] and biochemical studies [97] indicate that Lis1 binds to Nde1/Ndel1 at an adjacent region of the coiled-coil (Figure 3a). Interestingly, the C-terminus of Nde1 has been shown to fold back to the Lis1 binding site, suggesting a self-regulation mechanism of Nde1 [96,98].

The interactions between the IC SAH and p150^Glued^ CC1B are vital for the motility of dynein-1, as disrupting this interaction renders the motor complex inactive. For example, in the presence of a high concentration of Nde1/Ndel1, the motility of the DDA complex is abolished [95,96], presumably because Nde1/Ndel1 displaces p150^Glued^ CC1B from the IC. However, the precise mechanism by which disrupting the binding between p150^Glued^ CC1B and the IC SAH leads to dynein inactivation remains somewhat mysterious. One hypothesis is that p150^Glued^ provides additional rigidity by pulling on the IC via CC1B. Disengaging this interaction might make dynein too flexible for processive motion. However, in cryo-EM structures of the full DDA complex, although the LC Roadblock is visibly trailing behind the HCs, the interaction between p150^Glued^ and the dynein IC has not been observed [67,68,70], leaving the nature of this interaction and its importance for DDA complex assembly and motility unclear.

Collectively, the emerging evidence suggests a probable mechanism where the IC serves as a central hub, modulating the binding of dynein cofactors to activate dynein (Figure 3c). Upon Lis1 binding to Nde1/Ndel1, Nde1/Ndel1 facilitates Lis1’s recruitment to the autoinhibited dynein complex by interacting with the IC SAH. After this binding event, it is plausible that the interaction either weakens the affinity of Lis1 for Nde1/Ndel1, given their proximity at the binding sites, or Lis1 simply exhibits a higher affinity for the motor domains. In either case, Lis1 detaches from Nde1/Ndel1 and instead interacts with the nearby dynein motor domains, acting like a wedge between them [99]. This interaction transforms the autoinhibited dynein into a conformation that is more amenable to assembling with dynactin and an adaptor.

At this stage, either the CC1B domain of p150^Glued^ displaces Nde1/Ndel1 from IC due to its higher affinity for the IC, bringing dynactin closer to dynein to form the DDA complex (Figure 3c, bottom right); alternatively, the DDA complex assembles independently of this interaction, and after assembly, p150^Glued^’s CC1B displaces Nde1/Ndel1 from the IC SAH (Figure 3c, bottom left). Regardless of the pathway, by this stage, the complex is fully assembled and ready to move along MTs.

Besides the role of activating dynein, the IC also directly modulates dynein’s mobility. For example, in *Saccharomyces cerevisiae*, the dynein HC forms a dimer and moves processively alone MTs even in the absence of the IC [86]. However, the run length of dynein is significantly reduced without the IC [86]. Similarly in mice, a mutation in the HC impacting its binding to IC leads to a “legs at odd angles” (*Loa*) phenotype [100]. This dynein-1 mutant shows decreased processivity at both the single-molecule [100,101] and cargo-transport level [100], possibly due to a higher tendency for side-stepping on MTs [100]. These findings collectively suggest that the IC’s “clamping” of the HCs together is important for coordinating the forward movement of the motor domains.

#### 2.2.3. Diversity of IC

In humans and mice, there are two homologs of the IC for dynein-1: IC1 and IC2. IC2 is ubiquitously expressed in cells [102], while IC1 is primarily expressed in the brain [103]. Each of these homologs has at least three splicing isoforms [75,104]. These isoforms have been shown to form both homodimers and heterodimers when overexpressed in cells [105], and they are capable of binding to all LCs [88]. In *Drosophila melanogaster*, which has only one IC for dynein-1, at least 10 alternatively spliced IC isoforms are used in a tissue-specific manner [106]. Additionally, post-translational modification of IC, such as phosphorylation, could modulate IC’s selectivity for either Nde1 or p150^Glued^ [79] or regulate its binding to p150^Glued^ [107]. These mechanisms significantly contribute to the diversity of IC functions (reviewed in [108]).

As the primary cytoplasmic retrograde transport motor, dynein-1 performs a surprisingly wide range of functions. Diversification of the non-catalytic subunits can effectively expand the functions of dynein-1 in a modular manner. However, the specific functions of the different homologs and isoforms, as well as the regulatory roles of post-translational modifications, remain underexplored. This is partly due to the complexity and essential, varied roles of dynein in mammalian cells. The precise ways in which different isoforms fine-tune dynein-1’s function in different tissues, and the role of the IDRs in regulating dynein when the isoforms do not alter LC binding, are still unclear. Given the presence of a disordered region of approximately 100 amino acids between the LC8 and Roadblock binding sites (Figure 4a), it would be interesting to investigate whether other binding partners interact with the IC in this region.

### 2.3. Dynein-1 Light Intermediate Chain (LIC)

#### 2.3.1. Structure of LIC

The LIC of dynein-1 is a RAS-like protein with a molecular weight of 50–60 kDa [109]. It is characterized by an ordered N-terminal Ras-like globular domain and a disordered C-terminal domain that facilitates interactions with various adaptors. The Ras-like domain forms a crucial association with the HC through a patch of aromatic residues [109]. This interaction stabilizes the HC and prevents aggregation, likely by shielding an aromatic patch on the HC through LIC binding [59,110].

Although the human LIC Ras-domain retains the ability to bind to GTP, it has lost its capability of hydrolysis, in contrast to the fungus *Chaetomium thermophilum* LIC, which lacks a GTP-binding pocket entirely [109]. While GTP binding might induce conformational changes in LIC and regulate its interaction with the HC, it is more likely that nucleotide binding primarily provides structural support for the Ras-domain [109].

LIC adopts a distinctive crescent shape on the HC tail region [68]. Subsequent cryo-EM structures unveil that the globular domain of LIC binds at HB 6 of the HC tail [7], with two additional extended intensities protruding from the globular domain above and below, making contacts along the HC at HB 5 and 7 [7] (Figure 5a). This confirms an earlier biochemical study showing overlapping binding regions of IC and LIC on the HC [111]. As a result of this extensive binding, LIC, in coordination with IC, forms a supportive scaffold that effectively sandwiches the HC, providing essential rigidity and stability to the dynein complex (Figure 5a).

In contrast to the structured N-terminus, the C-terminus of LIC appears mostly disordered, featuring interspersed short helixes. This aspect is discussed in the subsequent section.

#### 2.3.2. Function of LIC

The C-terminus of LIC is predominantly disordered, interspersed by two short helices, helix-1 and helix-2 [113,114] (Figure 4b). Numerous studies have indicated that helix-1 interacts with multiple dynein-1 adaptors [113,114,115]. Helix-1 is a highly conserved amphipathic helix across species, showcasing remarkable versatility in binding to various protein folds, such as the Hook domain of the Hook family, CC1-Box domain of the BicD family, and EF-hand in the calcium-binding family in a 2:2 ratio (Figure 5b) [113].

Helix-1 utilizes a highly conserved hydrophobic surface to establish interactions with adaptors. For instance, one flexible helix (helix-8) in the Hook domain is capable of adopting different conformations [116], and helix-1 of LIC shifts the equilibrium and induces the conformational change in the Hook domain’s helix-8 from a straight form to a V-shape through hydrophobic interactions [113]. Mutagenesis studies further support the notion that the flexible helix-8 in the Hook domain is the primary binding site for LIC [116].

Similarly, helix-1 uses the same hydrophobic surface to interact with BicD2 and CRACR2A [115]. ColabFold predictions are consistent with these findings, suggesting interactions between LIC1 helix-1 and BicD2, JIP3, Hook1, and CRACR2A, as depicted in Figure 5b. The cryo-EM structure of the DDR complex provides an additional validation for these interactions, revealing that the region following the globular Ras-like domain in LIC extends along the HC tail HB5, reaching for the BicDR1 adaptor with helix-1 (Figure 5a) [7].

The interactions between LIC and adaptors are pivotal for the motility and functions of dynein along the endosome–lysosome pathway. For example, Kazrin interacts with dynein LIC and dynactin, facilitating the recruitment of the complex to early endosomes [117]. Similarly, Hook1 and Hook3 are involved in early endosome processes [118,119]. Disrupting the interactions between LIC helix-1 and Hook adaptors impairs the motility of the dynein–dynactin–Hook (DDH) complex in vitro [113,115,116]. Possibly downstream to Kazrin, FIP3 serves as a link between dynein and recycling endosomes through its interaction with LIC [120]. In the context of late endosomes and lysosomes, Rab7-interacting lysosomal protein (RILP) interacts with LIC, playing a crucial role in recruiting dynein-1 to these organelles [121]. Disrupting critical hydrophobic residues in LIC helix-1 has been shown to impair lysosome transport in vivo, highlighting the physiological importance of these interactions [113].

Dynein-1 plays indispensable roles in cell division, particularly during mitosis, where it localizes at the spindle poles, kinetochore, and cell cortex [1]. In these mitotic structures, LIC assumes critical functions in localizing dynein-1 to the correct locations. Pericentrin, a crucial key component for MT organization [122], is known to localize at the centrosome [123] and exhibits colocalization with LIC [124]. However, unlike the canonical dynein-1 adaptors which interact with dynactin, dynein HC, and dynein LIC, pericentrin solely interacts with LIC [124]. Biochemical studies have pinpointed the specific interaction site between pericentrin and LIC within the region of amino acids 140–236 of LIC1 [125]. Structural studies reveal that dynein HC interacts with LIC on the opposite interface [6,7,67], suggesting possible concurrent interactions between pericentrin and the DDA complex with LIC. At kinetochore, the Rod–Zw10–Zwilch (RZZ) complex recruits adaptor Spindly, which subsequently recruits dynein via the LIC and dynactin to the kinetochore [126]. A recent study indicated that LIC, while interacting with Spindly, can also recruit pericentrin, which then attracts the γ-tubulin ring complex, promoting MT nucleation at the kinetochore [127]. At the cell cortex, the nuclear mitotic apparatus (NuMA) anchors at the cortical region and facilitates mitotic spindle positioning by recruiting dynein-1 and dynactin to capture astral MTs [51,128]. It interacts with LIC via its N-terminus [50,129], which contains both a Hook domain and a CC1-Box domain [129].

Dynein-1 also drives chromosomal movements in the prophase I of meiosis [130]. It links to the chromosomes via the Linker of nucleoplasm and cytoplasm (LINC) complexes [131], which contain the transmembrane protein KASH [132]. Recent studies have demonstrated that KASH5, an adaptor, binds to dynein-1 LIC and activates dynein-1 in the same fashion as other adaptors, utilizing its EF-hand domain, which is not regulated by calcium [133,134].

Besides interacting with pericentrin, the Ras-like domain of LIC also interacts with the neighboring dynein HC motor domain (AAA2 and AAA3) within the DDR complex that contains two dyneins [67]. This potentially helps to synchronize the two dyneins in the DDR complex.

#### 2.3.3. Diversity of LIC

In mammalian dynein-1, there are two homologs of LIC, namely LIC1 and LIC2 [26,135], sharing 65% sequence identity. Despite this homology, they do not coexist in the same dynein-1 complex [125], resulting in distinct dynein subpopulations that have non-overlapping roles in cellular functions. For instance, during mitosis, LIC1 predominantly localizes at the kinetochore from metaphase to anaphase [136], playing a pivotal role in removing spindle-assembly checkpoint (SAC) components from kinetochores [137]. In contrast, LIC2 is concentrated at the spindle poles throughout the entire mitotic process [136], proving essential for mitotic spindle orientation [138,139]. However, a later study suggests that LIC2 also participates in removing SAC components, exhibiting a stronger and more diverse role [140].

At the molecular level, although an NMR study has revealed structural similarities in the C-terminal helices of LIC1 and LIC2 [141], these homologs can exhibit a different affinity for adaptors. For example, LIC1 has a stronger affinity for BicD2 than LIC2 [142]. As a result, LIC1, but not LIC2, is crucial for BicD2-mediated interkinetic nuclear migration (INM) [142], a process that is essential for the proliferation of embryonic neural stem cells (radial glial progenitors) [143]. LIC1 contains an extended flexible linker between helix-1 and helix-2 that is absent in LIC2. This extended linker in LIC1 might enhance the accessibility of helix-2 to binding partners located further away from the LIC core, although the function of helix-2 remains unknown at present. A study in *Caenorhabditis elegans* demonstrated that deletion of helix-2 overall does not affect dynein function [114].

Unlike IC, the diversity of LIC is not augmented by alternative splicing; to date, no known isoforms exist. However, post-translational modifications significantly increase the complexity of LIC. The importance of how the post-translational modifications of LIC contribute to the diverse functions of dynein has been recently explored and postulated in a comprehensive review [144].

### 2.4. Dynein-1 Light Chains (LCs)

Mammalian dynein-1 has three different LCs (Figure 6a–c), each with two homologs: Roadblock (DLRB1 and DLRB2), LC8 (DYL1 and DYL2), and Tctex (DYLT1 and DYLT3). They bind the IC in a 2:2 ratio, forming the IC-LC subcomplex.

#### 2.4.1. Roadblock

Roadblock, or LC7, which was first identified in both *Drosophila* and *Chlamydomonas* [149], is a ~11 kDa protein that dimerizes on its own (Figure 6a), as shown by both X-ray and NMR studies [90,145,150]. An early study identified the binding site of Roadblock on IC, located downstream of IC’s splicing sites and upstream of the WD repeats, suggesting no preference for specific IC isoforms [84]. The crystal structure of Roadblock bound to a short peptide of IC suggests that it converts the intrinsically disordered peptide of IC into a more ordered conformation [90]. Subsequent cryo-EM structures of both autoinhibited [6] and activated dynein-1 [7,67] confirm that the dimeric Roadblock binds near the IC’s WD repeats (Figure 2a). This binding arrangement effectively positions the two IC WD repeats closely, indicating that the primary role of Roadblock is to function as a clamp, ensuring IC’s association and thereby maintaining proximity of the HC tails. Notably, dynein-1 in *Saccharomyces* lacks a homolog equivalent to Roadblock [149], raising questions about whether yeast IC has evolved an alternate mechanism for maintaining IC proximity. In the filamentous fungus *Aspergillus nidulans*, which has a Roadblock homolog (RobA), Roadblock deletion leads to a mild phenotype [151], while homozygous Roadblock-1 null mice are not viable [152], suggesting that yeast may have, indeed, evolved a compensatory mechanism.

There are two vertebrate homologs of Roadblock: Roadblock-1 and Roadblock-2, sharing 75% sequence identity. Single-molecule studies show that DDB complexes display similar motility in vitro with either homolog [153]. Nevertheless, Roadblock-1 knockout is embryonically lethal, underscoring the non-redundant cellular functions of the homologs [152]. Recent research has shed light on their distinct roles: while Roadblock-1 is ubiquitously expressed in mouse tissues and plays a crucial role in ensuring the integrity of the mitotic spindle pole, Roadblock-2 is exclusively involved in meiosis [153]. In meiotic cells, Roadblock-2 targets NuMA to the spindle pole, and its absence leads to spindle pole defects, including multipolarity and misalignment [153].

#### 2.4.2. LC8

LC8 is a ~10 kDa protein that is highly conserved across eukaryotic cells [26]. Much like Roadblock, LC8 forms dimers [154] (Figure 6b), which undergo dissociation at low pH conditions due to the ionization of a histidine at the dimerization interface [155]. It dimerizes with the short disordered region in the N-terminus of IC at a 2:2 ratio (Figure 2a), enhancing the structural order of IC [83].

While LC8’s role in facilitating the dimerization of the IC is well-established, its potential involvement in other regulatory functions within dynein-1 remains largely unknown. One study indicated its collaboration with ADP-ribosylation factor-like 3 (Arl3) in dissociating dynactin from dynein. In this process, LC8 binds to the IC, and Arl3 binds to the linker region between CC1 and CC2 of p150^Gluded^ [156]. The precise mechanism of this disassembly, however, remains elusive.

In *Saccharomyces*, dynein-1 features only LC8 (Dyn2) as its light chain, with two LC8 dimers binding to the N-terminal region of the IC. The absence of LC8 impedes the IC’s binding to the HC, likely due to LC8’s role in promoting IC dimerization and enhancing its affinity for the HC [86]. Notably, DYN2 is one of the few genes in *Saccharomyces* with two introns and undergoes alternative splicing [157], yet the impact of LC8 isoforms on yeast dynein’s function is unclear. In mammals, there are two homologs of LC8, LC8-1 and LC8-2, differing by only a few residues. Thermodynamic studies suggest subtle differences in their binding to partners [158], but distinct functionalities of these LC8 variants have not been established.

Initially identified as a subunit of axonemal dynein [159,160], LC8 was later discovered to also exist in cytoplasmic dyneins [161,162,163]. Since then, LC8 has gained recognition as a versatile protein known for its interaction with a diverse range of proteins, often characterized by a threonine–glutamine–threonine (TQT) motif. This has established LC8 as a central hub for dimerization [158,164,165]. Its binding partners span a wide spectrum of proteins, including Nup159 in the nuclear pore complex (NPC) [166], myosin-Va [167], p53 and MRE11 in DNA double-strand break response [168,169,170], and various viruses [171,172]. This remarkable diversity underscores LC8’s involvement in functions beyond dynein. LC8’s essential functions in metazoans and its presence in plants despite their lacking dynein complexes suggest a broader role for this small protein [158].

#### 2.4.3. Tctex

Tctex was initially identified as a subunit in mouse brain dynein-1 [173] and subsequently in axonemal dyneins [174,175]. This approximately 11 kDa protein forms a dimer [147] (Figure 6c) and binds adjacent to LC8 on IC [88] (Figure 2a). The binding of Tctex to IC enhances LC8 binding through avidity [88]. Although Tctex and LC8 share structural similarities (Figure 6b,c), they lack sequence similarities and likely have different evolutionary origins [147].

In addition to binding to IC, Tctex-1 directly interacts with rhodopsin, a protein responsible for low light sensing [176]. Mutagenesis studies mimicking phosphorylation/dephosphorylation indicate that the post-translational modification regulates the binding of Tctex-1 to rhodopsin [177]. Tctex and Roadblock also interact with unc104 in *Caenorhabditis*, a highly processive and fast kinesin that travels towards the plus-end of MTs, thereby facilitating bidirectional cargo transport [178].

Mammalian dynein-1 has two Tctex homologs, Tctex-1 and Tctex-3, which are mutually exclusive in their binding to dynein complexes, suggesting that they form only homodimers [179]. Additionally, their cargo-binding preferences differ, with Tctex-1 associating with rhodopsin, while Tctex-3 does not bind to rhodopsin [180].

Like LC8, Tctex functions as a dimerization hub independent of dynein [181]. Intriguingly, Tda2 in *Saccharomyces*, which is structurally homologous to Tctex [182], does not associate with dynein-1 but is involved in actin assembly [181,182]. In vertebrates, Tctex also plays a dynein-independent role in regulating actin dynamics [183,184,185]. This suggests that Tctex may predate dynein’s evolution and was repurposed as a subunit during evolution. While mammalian dyneins include Tctex as a subunit, yeast dynein has a diverged evolutionary, leading to Tctex’s loss as a subunit.

#### 2.4.4. Summary of LCs

All the light chains (LCs) discussed in this section—Roadblock, LC8, and Tctex—are shared among cytoplasmic dynein-1, cytoplasmic dynein-2, and axonemal dyneins. The small size of LCs might contribute to their broad binding pattern, given their limited binding interfaces with other proteins, as evidenced by the µM range of binding affinity observed for LC8 and Tctex with IC peptides. Despite their promiscuity, these LCs bind to specific sites on dynein IC without interchanging. It remains unclear why mammalian dynein requires three different LCs, whereas *Saccharomyces* cytoplasmic dynein functions with only one LC. It is plausible that the diversity of LCs corresponds to the array of functions required by mammalian dynein, which must be versatile for numerous cellular functions. Strategies such as gene duplication, alternative splicing isoforms, and post-translational modifications are employed by IC, LIC, and LCs to increase subunit diversity. Dynein further enhances this diversity by incorporating various types of LCs to interact with different cellular components. Several questions about LCs remain unanswered, including how their diversity affects dynein-1’s functionality; whether all LCs are necessary for assembling a dynein complex, or a subset of light chains is sufficient; and how they might impact dynein’s motility and function.

## 3. Cytoplasmic Dynein-2

The cilium, or flagellum, is a specialized membrane-bound organelle that protrudes from eukaryotic cells, enabling the cells to sense and navigate their environment [186]. It contains a central structure, the axoneme, consisting of microtubule doublets (MTDs). There are two types of cilia, motile cilia and immotile (primary) cilia. Motile cilia typically have nine MTDs surrounding two central single MTs (9 + 2 arrangement) and contain axonemal dyneins, enabling them to beat [44]. In contrast, primary cilia lack the central MT pair and axonemal dyneins (9 + 0 arrangement), functioning as a sensory unit [187] (Figure 7, bottom panel left). For cilia to exchange information and material with the cell body, intraflagellar transport (IFT) carries cargoes and components along the axoneme tracks [188,189] (Figure 7, bottom panel). These large IFT “trains” of protein complexes move along the outer surface of the axoneme, with the transition zone (TZ) at the base of the axoneme serving as a checkpoint, regulating the passage of proteins in and out of the cilium [190,191]. Molecular motors drive the movement of IFT trains: kinesin-2 family members facilitate anterograde transport, while cytoplasmic dynein-2 (referred to as dynein-2, also known as IFT dynein and dynein-1b in *Chlamydomonas*) powers retrograde transport [192]. A comprehensive study using CLAM and cryo-EM showed that these motors move along different tubules of the MTD tracks, with kinesin-2 using the B-tubule and dynein-2 using the A-tubule to minimize potential traffic jams [193]. This is further supported by another study using cryo-EM and U-ExM, indicating that IFT trains are loaded onto the B-tubule via kinesin-2 at the TZ [194]. Structural details about retrograde trains are less known, presumably due to their less rigid and more heterogeneous nature, posing challenges for cryo-EM studies. Besides the motors, IFT trains include components such as the IFT-A complex [195,196,197,198], IFT-B complex [199], and BBSome [200,201,202,203].

### 3.1. The Structural Architecture of Dynein-2

The composition of dynein-2 is similar to dynein-1, containing HC, IC, LIC, and LCs [163]. Nevertheless, several specific features distinguish the associated subunits of dynein-2 from those of dynein-1: instead of a homodimer, dynein-2 has two different ICs that form a heterodimer in the presence of LCs; dynein-2 binds more LCs to the IC (Figure 2b), and it has a distinct LC Tctex homolog that is absent in dynein-1.

Similar to dynein-1, the dynein-2 complex, when isolated, assumes an autoinhibited conformation. In this state, its two motor domains face away from each other, and the stalks cross to prevent MT binding [8,205] (Figure 1b). The HC tail adopts an asymmetrical configuration, featuring one twisted and one straight HC, only with the twisted HC more zig-zagged than in dynein-1, due to the constrain imposed by the heterodimeric ICs (Figure 1a,b) [8]. In addition, the subunits in dynein-2 make more intermolecular contacts than in dynein-1 [8].

In comparison to dynein-1, less is known about the active state of dynein-2. Dynein-2 does not require dynactin and adaptors to be active [163], suggesting that IFT complexes may play a role in its activation. Unlike dynein-1, where the mutation-induced opening of the complex does not activate it for motility [6], mutations in the linker domain are sufficient to activate dynein-2 in vitro [205]. This suggests a different activation mechanism for dynein-2. Additionally, in *Caenorhabditis*, these mutations lead to retrograde movements of dynein-2 in the absence of IFT-A, an essential component for retrograde IFT train assembly and the retrograde motility of the wild-type dynein-2 [206]. Nonetheless, the details of how dynein-2 is activated when bound to retrograde IFT trains and how it moves along the axoneme in teams remain unknown.

### 3.2. Dynein-2 and IFT Trains

IFT trains consist of IFT-A and IFT-B complexes. In anterograde trains, IFT-A is closer to the membrane of the cilium, while IFT-B is sandwiched between IFT-A and MTs. Dynein-2 primarily interacts with the IFT-B in anterograde trains. Structural studies have revealed that dynein-2 is loaded on the anterograde IFT trains in its autoinhibited conformation [108,109] and transported to the tip of the cilium via kinesin-2 (Figure 7, top), effectively preventing a tug-of-war between these two opposing motors. Two layers of inhibition of dynein-2 in an anterograde train are implemented. First, its linker and stalk are trapped in a restricted conformation by the stacking of the motor domains, preventing the dynein-2 motor domains from forming a parallel conformation [204,205]. Second, when bound to the IFT-B complex, dynein-2 HC assumes an upside-down arrangement, with the tail pointing towards the MTs, while the MTBD points away and binds into a negatively charged groove of IFT-B, further preventing interactions of the motor with MTs [204].

The zig-zagged conformation in the autoinhibited state of dynein-2 is tailored to bind to the IFT-B complexes, spanning 7-8 IFT-B repeats. This suggests the dynein-2 complexes bind to assembled IFT [8], supported by a cryo-EM study of the TZ and a detailed cryo-EM structure of the anterograde IFT train [194]. Interestingly, only the HC has shown extensive interactions with IFT-B in cry-EM structures, despite the non-catalytic subunits binding along the tail of the HC [198]. However, there may be flexible interactions between these subunits and the IFT complexes not visible in cryo-EM structures. Indeed, immunoprecipitation assays have revealed interactions between the IC of dynein-2 and the IFT-B complex [207,208], highlighting the dynamic nature of these molecular associations.

How the trains turn around at the tip of cilia and how dynein-2 is activated is largely unclear. While existing evidence indicates a level of fragmentation and a disassembly/reassembly process of the trains at the tip, regulated by Ca^2+^ [209], a recent study suggests that switching directions is an intrinsic property of the trains. When the train is blocked and derailed, they change directions without external regulators [210]. This suggests a potential mechanically induced conformational change mechanism, wherein compression of the train may trigger derailment and remodeling of IFT trains. Such a process could activate dynein-2 for IFT turnaround.

### 3.3. Dynein-2 Intermediate Chains (ICs)

Unlike the dynein-1 IC, which forms a homodimer, dynein-2 presents a unique configuration where two distinct ICs, WDR60 and WDR34 (initially identified in *Chlamydomonas* as FAP163 [211] and FAP133 [212], respectively), bind to the HC as a heterodimer [163]. Similar to the dynein-1 IC, both ICs of dynein-2 exhibit C-terminal WD repeats responsible for anchoring the ICs onto the HC and an N-terminal IDR that associates with the LCs. Unexpectedly, WDR60, even without the WD repeats, shows the ability to integrate into the IFT trains in *Caenorhabditis* [213], indicating that the N-terminus of WDR60 may engage in interactions with IFT complexes independently of the HC. WDR60 contains an extra ~450 amino acids at the N-terminus compared to WDR34 (Figure 4a). This region is highly charged and predicted to be disordered, potentially engaging in various protein–protein interactions with the IFT trains.

Both of the ICs are critical for the dynein-2 functions, with deficiencies in either causing developmental defects [208]. However, they have distinct roles in dynein-2: WDR-60 is important for dynein-2 assembly, recruitment onto the IFT train, and transition zone crossing [213]. Without WDR-60, less dynein-2 is incorporated into the anterograde IFT trains, reducing the available dynein-2 for retrograde transport. It is unclear if WDR-60 also directly affects dynein’s motility, such as playing a role in activating the dynein-2 complex. WDR-34 is important for the assembly and function of dynein-2 but has less of an impact on the loading of dynein-2 onto the IFT train [214]. Cryo-EM structures of the assembled dynein-2 complex show that WDR34 binding breaks the symmetry of the homodimeric HC, with one HC relatively straight and the other in a twisted zig-zag conformation, fitting into the contour of IFT-B complexes [8].

While the mechanism of how heterodimeric ICs are selectively integrated into the dynein-2 complex is unknown, it is reasonable to speculate that, in the absence of one IC, a homodimer might form due to the identical nature of their binding partners, the HCs and LCs. If a homodimer of either WDR60 or WDR34 binds to the HC, the dynein-2 complex might assume a more symmetrical shape, which would not fit well with the IFT-B complexes. Interestingly, *Caenorhabditis* has only one IC for dynein-2 [215]. A comparative analysis of the dynein-2 complex in *Caenorhabditis* and humans could yield more insights into how the IC shapes the conformation of dynein-2.

### 3.4. Dynein-2 Light Intermediate Chain (LIC)

Dynein-2 LIC (DYNC2LI1, or LIC3), first identified in mammalian cells [216], is a homolog of dynein-1 LIC, implying the presence of an N-terminal Ras-like domain [109], which was later confirmed by the dynein-2 structure [8]. In a pattern resembling dynein-1 HC and LIC interactions, LIC3 binds to dynein-2 HC’s HB 6, extending its two arms outward to engage adjacent HC helical bundles [8]. Within the HC tail, only the sequences of the LIC and IC binding sites are well conserved between dynein-1 and dynein-2 HC [217], highlighting the evolutionary connections and the significance of HC-LIC interactions. Even when HC disassembles into a monomeric state in the absence of IC-LC, it maintains its association with LIC [8]. Like dynein-1 LICs, LIC3 also interacts with the neighboring HC motor domain when dynein-2 is loaded onto the IFT-B complex [8]. This interaction presumably reinforces dynein-2’s binding to IFT-B complexes by stabilizing the dynein-2 assembly chain. LIC3 also interacts with a subunit of IFT-B complex, IFT-54 [207,218], potentially strengthening the interactions between dynein-2 and IFT-B complexes.

A feature unique to LIC3, compared to dynein-1 LICs, is the absence of the C-terminal disordered region (Figure 4b). This is notable because LIC1 and LIC2 use this region to interact with the N-termini of various adaptors, which dynein-2 does not rely on [163]. Unlike dynein-1, where LIC primarily contacts HC, LIC3 in dynein-2 makes contacts with other subunits besides the HC in the autoinhibited conformation, such as IC WDR60, LC Roadblock, and LC LC8, crucial for the structural stabilization of the entire complex [8]. Despite these insights into LIC3’s structure within the dynein-2 complex, the functional aspects of LICs in general remain unknown. While LIC3 is an obligatory binding partner for the HC and plays a crucial role in maintaining the stability of the dynein-2 complex [219], its precise roles and regulatory functions beyond this association are yet to be fully unveiled.

### 3.5. Dynein-2 Light Chains (LCs)

Dynein-2 contains all the LCs of dynein-1, along with an additional Tctex homolog, Tctex1D2, identified as an unique dynein-2 LC in human cells [163]. Unlike dynein-1, which has a homodimer of each LC, dynein-2 contains one homodimer of Roadblock, three homodimers of LC8, and one heterodimer of Tctex (Figure 2b). The dimeric Roadblock likely functions similarly to its role in dynein-1, by clamping the two ICs closely together. The consecutive LC8 dimers bind much closer to the WD repeats (Figure 2), in contrast to the ~20 nm distance between Roadblock and LC8 in dynein-1. The reason for the three dimers of LC8 is unknown. The functionality of the heterodimer Tctex also remains unclear, although the interaction between Tctex1D2 and the IC WDR60 is crucial for ciliary protein trafficking [220]. Interactions of IC WDR34 with the LCs are also required for ciliary protein trafficking [221]. Overall, the contribution of LCs to the functionality of dynein-2, particularly in its active state during retrograde transport, remains poorly understood.

## 4. Axonemal Dyneins

The axoneme, a highly complex and organized structure found in motile cilia, plays a crucial role in cellular locomotion and fluid flow on the surface of various human cells, including those in the nervous system [222], the respiratory tract [223], and the motility of sperm [224]. In motile cilia, a row of dyneins forms the outer dynein arms (ODAs) that anchor on the A-tubule of MTDs via docking complexes, and another row of dyneins forms the inner dynein arms (IDAs) [44,225], which also anchor on the A-tubule (Figure 8a). The dyneins dynamically bind to or dissociate from the B-tubule of neighboring MTDs depending on the nucleotide states of their motor domains. An extensive protein network, consisting of MT inner proteins (MIPs), resides within the lumen of the MTD tubules. Nexin–dynein regulator complexes (N-DRC) link neighboring MTDs and regulate ciliary motility, while radial spokes (RSs) connect MTDs to the central apparatus (CA)—the central two singlet MTs shrouded with protein complexes. Recent advances in cryo-EM microscopy and image processing techniques have significantly enriched our molecular understanding of axonemes (reviewed in [226]), revealing detailed architectures of the full axoneme [10,227] and its components, such as ODAs [9,29,45,228,229], MIPs [230,231,232], N-DRC [233], RSs [234,235,236], and CA [237,238].

The green algae *Chlamydomonas reinhardtii* [239,240] and the ciliate *Tetrahymena thermophila* [241] are two model organisms that are indispensable in axoneme research, providing invaluable insights. However, it is noteworthy that, while the axonemal structure maintains overall conservation across species, there are variations in composition and local structures among axonemes from different organisms [10]. In particular, *Chlamydomonas* and *Tetrahymena* exhibit a more complex composition of axonemal dyneins compared to the human epithelial cilia dyneins, with additional ICs and LCs [10]. In mammals, even within the same species, respiratory cilia and sperm flagella display different structural elements [242]. The following sections focus on discoveries based on these model systems, complemented by insights derived from human axonemal dynein where applicable.

### 4.1. The Assembly of Axonemal Dyneins

Distinct from cytoplasmic dyneins, which form homodimers with two identical HCs, axonemal dyneins exhibit a remarkable diversity. Multiple dynein HCs, each encoded by a unique gene, come together to form single-headed, double-headed, or triple-headed complexes [44]. For the outer-arm dyneins (OADs), the dynein HCs form either heterodimers or heterotrimers, depending on species, arranged in 24 nm intervals. Both *Chlamydomonas* and *Tetrahymena* possess triple-headed OADs (Figure 1e), while the human axoneme has double-headed OADs (Figure 8b, right), lacking the outermost HC. The innermost two HCs of the triple-headed HCs and the double-headed HCs are bound by heterodimeric ICs and a substantial number of LCs, which interact with either ICs or HCs. Similar to the cytoplasmic ICs, the ICs of OADs also have a disordered region at the N-terminus that provides the scaffold for LC binding (Figure 2c). The outermost HC of the triple-headed OADs is a diverged paralog of the middle HC [27], featuring an N-terminal γ-kelch domain that latches onto the tail region of the middle HC [9,228,229]. It does not have associated IC; in *Chlamydomonas*, an LC directly interacts with this HC. Notably, before OADs are transported and localized to axonemes, they are tightly packed and inhibited in a conformation similar to cytoplasmic dyneins by a protein named Shulin [45] (Figure 1c).

For the inner-arm dyneins (IADs), six single-headed dyneins (IAD*a–e* and IAD*g*) and one heterodimeric complex (IAD*f*) with its IC-LC tower, structurally identical to the ODAs (Figure 8b, left), bind to the A-tubule via actin and LCs [10]. Together, they are arranged at 96 nm intervals. In *Chlamydomonas*, each HC is encoded by a unique gene, whereas in humans, IAD*b* and IAD*e* share the same HC [10]. In addition to the ICs docking on the HCs, another IC—IC97 in *Chlamydomonas* or DNAI7 (Las1) in humans—interacts with the LCs on the IC-LC tower [10]. Moreover, *Chlamydomonas* has an additional protein, FAP120, in the vicinity of IC97 and the IC-LC tower [10].

### 4.2. Axonemal Dynein Intermediate Chains (ICs)

The OADs assume a tail-to-head compact conformation, in which the tails of one OAD complex stack onto the motor heads of the adjacent complex [9,228,229]. This conformation ensures the sequential activation of the OADs, with the downstream OAD inhibited by the upstream OAD unless it undergoes a conformational change after ATP hydrolysis. Both the WD repeats of one of the ICs and the tail of the HC participate in the interactions between the OADs. In contrast to the dynein-1’s IC-LC tower, which trails behind the HC in its activated form, the IC-LC tower of the OAD closely associates with the HC. Furthermore, instead of symmetrically binding in the middle of the HC, it rotates towards the inner side, facing the IADs.

The IC-LC tower of IAD*f* has a similar structure to that of the OADs, associating asymmetrically with HCs on one side. The IC on the other side binds to OAD’s LC Tctex [10]. This connection is likely crucial for the communication between the OADs and IADs. In addition, IAD*f* also contains a distinct non-WD repeat IC, IC97 [243], the functional role of which remains unknown.

### 4.3. Axonemal Dyneins Lack Light Intermediate Chain

In contrast to the cytoplasmic dyneins, axonemal dyneins do not have LICs [244]. This absence suggests the crucial role of LICs in providing rigidity for cytoplasmic dynein to move along MTs over long distances. The restriction of flexibility appears to be a common theme in activating and enhancing dynein movements. The interaction between LIC and adaptor [67,114,115,116], along with the recruitment of a second adaptor to a DDA complex on MTs [67], reinforces the rigidity of the DDA complex, with the former being particularly critical for dynein-1 motility (see Section 2.3).

Intriguingly, although lacking LIC, both OADs and IAD*f* have various elements binding to the region on HC corresponding to the LIC-binding sites on cytoplasmic dynein HCs, namely the HB 6 on the tail region. This indicates that binding at this region can modulate dynein’s behavior through diverse mechanisms. For example, in the OADs of *Chlamydomonas* and *Tetrahymena*, the N-terminal γ-kelch domain of the outermost OAD HC latches onto the HB 6 in the tail region of the middle OAD HC and assists the remodeling of the OAD array during ATP hydrolysis [9,228,229]. Additionally, the light chain LC4 engages with the HB 6 of the innermost OAD HC, serving as a regulatory element for OAD in response to calcium, which is further discussed in the following section.

### 4.4. Axonemal Dynein Light Chains (LCs)

In humans, both OAD and IAD*f* share the same set of IC-binding LCs, as well as a similar structural arrangement of the IC-LC tower, with dynein-2. From the N-terminus to the C-terminus of IC, the IC-LC tower comprises a Tctex heterodimer, three LC8 dimers, and a Roadblock heterodimer [9,10,45]. OADs in *Chlamydomonas* and *Tetrahymena* have additional LC8-like proteins in place of some of the LC8 [9,10,45], while OADs in humans feature a distinct LC8-type DNAL4 that remains under-characterized [43]. Despite the similarity, there are structural differences between the axonemal dyneins and dynein-2. While the Tctex dimer trails underneath LC8 in dynein-2 as a weak density in the cryo-EM structure [8], the Tctex dimer in both OADs and IAD*f* exhibits a distinctive bend back towards the adjacent LC8 due to constraints imposed by the N-terminal helical bar and the β-hairpin structure of the IC [9,10]. In the case of OADs, the unique bend back of the Tctex dimer in OADs facilitates its extension toward IAD*f*, establishing contact with the IC140 and HC tail of IAD*f*, functioning as one of the communication pathways between OADs and IADs [10]. In the case of IAD*f*, Tctex-2b is critical in the assembly of Tctex-1, IC97, and FAP120 of IAD*f* [245] in *Chlamydomonas*, as supported by the structure showing Tctex-2b bridging between IC97 and Tctex-1 [10].

Besides the LCs on the IC-LC tower, there are OAD-specific LCs that directly interact with HCs. Leucine-rich repeat protein LC1 (Figure 6d), which assumes a cylindrical shape as revealed by the NMR solution structure [246], is a light chain that binds directly to the MTBD of the innermost OAD (γ-HC in *Chlamydomonas* and α-HC in *Tetrahymena*) [247]. Structural studies reveal that LC1 has extensive interactions with the MTBD of the HC without directly interacting with MTDs, except for electrostatic interactions with the highly negatively charged β-tubulin C-terminal tail [9,248]. It is speculated that the LC1 can assist the MTBD in sensing the curvature of the MTD surface, leading to a rotary movement [248]. Furthermore, due to the binding of the LC1, the LC1-MTBD requires a wider inter-protofilament interface, guiding the OADs to localize to the correct positions on the MTDs [9]. A mutagenesis study demonstrated that LC1 is also crucial for the cytoplasmic preassembly of OADs and ciliary stability [249], although the molecular mechanism remains unclear.

LC3 (LC3BL in *Tetrahymena* and TXND6 in humans), a thioredoxin-like protein, binds to the tail region of the middle OAD in *Chlamydomonas* and *Tetrahymena* or the outermost OAD in humans. The thioredoxin superfamily is an ancient protein family that undergoes redox chemistry by utilizing a -Cys-X-X-Cys- motif (X represents any residue) [250]. It has been demonstrated that the redox state modulates *Chlamydomonas* flagellar beating patterns in vivo [251], controls the sign of phototaxis driven by differential motility of the two flagella [251,252], and alters the ATPase activity of its OAD HC’s in vitro [253]. Indeed, LC3 [254], LC3BL, and TXND6 all retain this motif (Figure 6e). Structurally, LC3 changes interactions with the HCs in OAD in response to the ATP hydrolysis of the motor heads [9]. Despite the advances in understanding LC3, it is not clear how the redox chemistry of LC3 modulates the activity of axonemal dyneins. LC5 is another thioredoxin-like protein that binds to the outermost OAD’s HC (a-HC) in *Chlamydomonas* [10,254]. It is missing in both *Tetrahymena* and humans. Interestingly, the human homolog of LC3, TXND6, has an additional nucleoside diphosphate kinase (NDPK) domain following the thioredoxin domain. NDPK is a type of enzyme that transfers γ-phosphate from a nucleoside triphosphate (NTP) to a nucleoside diphosphate (NDP), using a conserved histidine residue [255]. The NDPK domain in TXND6 retains this conserved histidine (Figure 6e). The function of this domain is unknown.

LC4, a calmodulin protein containing several EF-hand motifs, at least one of which binds Ca^2+^ [256] (Figure 6f), binds between the innermost OAD’s HC and the IC-LC complex in *Chlamydomonas* [10] and *Tetrahymena* [9]. It binds the tail of the HC and interacts with the IC-LC tower, using the opposite interface. In the presence of Ca^2+^, LC4 bends the tail of the innermost OAD HC, coming close to the ICs [257]; however, the implications of these conformational changes for the regulation of ciliary beating are not yet clear.

In the IADs, p28 is an LC that associates with the N-terminal domain of either a specific subset (in *Chlamydomonas*) [258] or all (in humans, IAD*d* and IAD*g* are amalgamated into one complex) [10] of the single-headed IAD HCs. Together with actin, the dimeric p28 anchors the HCs to the MTDs [10].

In summary, axonemal dyneins have an impressively large repertoire of LCs that fill the spaces among the HCs, with their functions largely unknown. Further understanding of how they contribute to the regulation of the ciliary beating would be of great interest.

## 5. Conclusions

Dynein is an ancient protein complex that likely existed and diverged into the three dynein families before the last eukaryotic common ancestor (LECA). This hypothesis is supported by the highly conserved sequences of HCs and ICs across species that harbor dynein [26]. Throughout evolution, most plants lost the dynein branch completely, while various animal cells have lost different branches of dynein depending on the functionality required. Serving as the primary motor for MT minus-end-directed movements, dyneins play an indispensable role in numerous cellular functions. Mutations in dyneins lead to a broad spectrum of human diseases [101,259,260,261]. Consequently, unraveling the molecular mechanisms of dyneins is crucial for a comprehensive understanding of their significance.

Recent advances in biochemical, single-molecule, and structural research have significantly enhanced our understanding of how dynein motor complexes assemble and how dynein generates motion and force. However, the specific contributions and regulatory functions of non-catalytic subunits in dynein’s operation are not yet fully understood. This review highlights their roles in maintaining the integrity of dynein complexes; providing structural support; regulating the enzymatic activity of the HC; directing the complexes’ cellular localizations; mediating interactions with cofactors or other proteins; and expanding the dynein interactome through homologs, isoforms, and post-translational modifications. Despite these insights, many questions about these subunits remain unanswered. Key inquiries include the role of IDRs of ICs in dynein’s functionality, the mechanisms by which LICs selectively target dyneins to specific cellular locations, the reason behind dynein’s multitude of LCs, and how these LCs modulate dynein’s diverse functions. Addressing these questions will deepen our understanding of dynein’s cellular functions and shed light on how mutations in dyneins contribute to human diseases.

## Figures and Tables

**Figure 1 cells-13-00330-f001:**
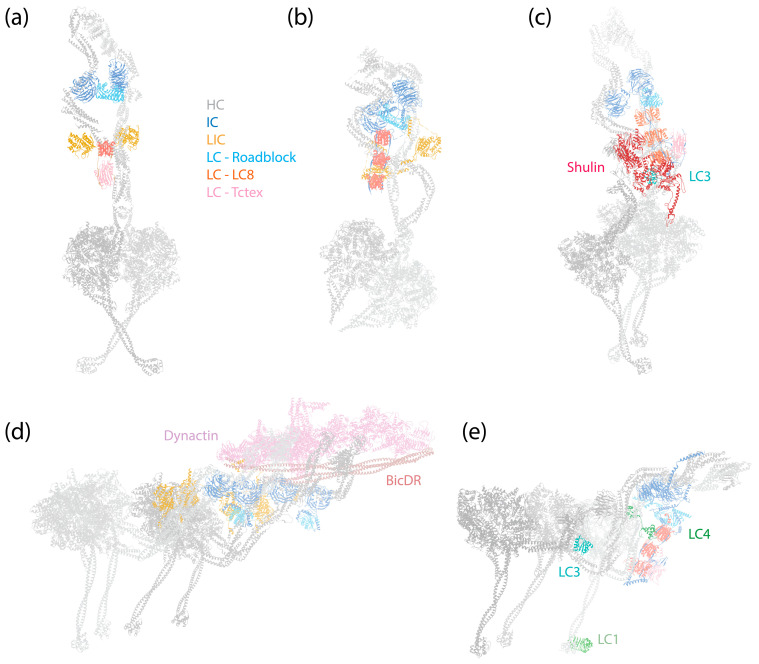
Structures of dynein complexes. Subunits are denoted by colored text. (**a**) Human cytoplasmic dynein-1 complex in an autoinhibited conformation [6] (PDB 5NVU). (**b**) Human cytoplasmic dynein-2 complex in an autoinhibited conformation [8] (PDB 6SC2). (**c**) *Tetrahymena* axonemal outer-arm dynein complex in an inhibited conformation imposed by Shulin [45] (PDB 6ZYW). (**d**) Human dynein–dynactin–BicDR1 complex with two dyneins and two BicDR1 adaptors [67] (PDB 7Z8F). (**e**) *Tetrahymena* axonemal outer-arm dynein complex bound to MTs in one of the two MT-binding states, with the central HC’s MTBD aligned with the outermost HC’s MTBD [9] (PDB 7KEK).

**Figure 2 cells-13-00330-f002:**
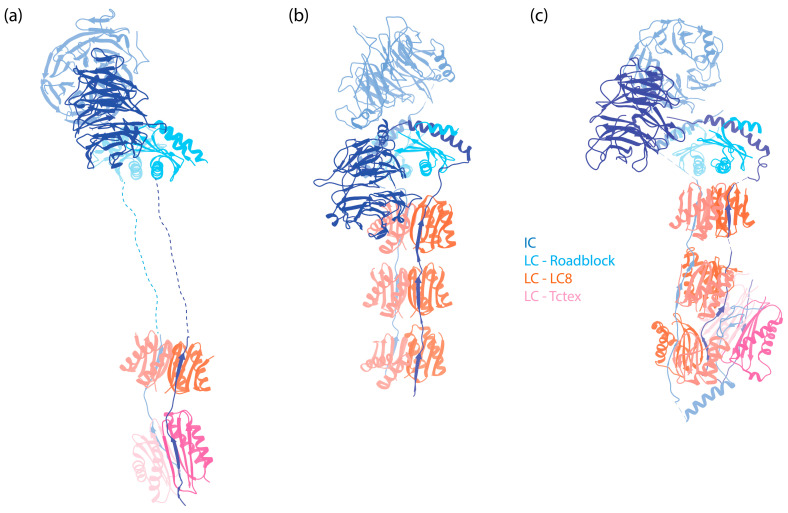
The IC-LC subcomplex in autoinhibited dynein complexes. The subunits are denoted by colored text. The structures are aligned horizontally at the LC Roadblock (in light blue) to allow for comparison of different dyneins. Notably, while LC8 and Tctex are approximately 20 nm away from the WD repeats of the IC in dynein-1, they are positioned much closer to the WD repeats in both dynein-2 and axonemal OAD. (**a**) IC-LC subcomplex of human dynein-1 [6] (PDB 5NVU). (**b**) IC-LC subcomplex of human dynein-2 [8] (PDB 6SC2). (**c**) IC-LC subcomplex of *Tetrahymena* OAD [45] (PDB 6ZYW).

**Figure 3 cells-13-00330-f003:**
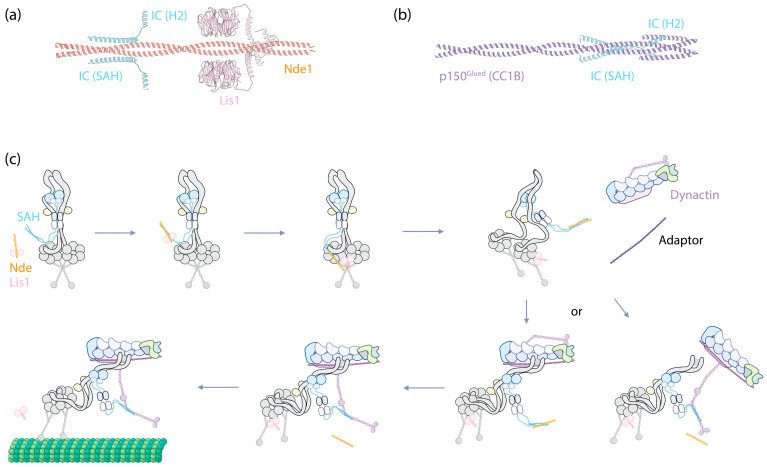
IC as the central hub for Nde(Nde1/Ndel1)/Lis1-mediated dynein-1 activation. (**a**) The structure of IC2’s SAH and H2 (Uniprot Q13409, amino acids 1–67) predicated by ColabFold to interact with Nde1 (Uniprot Q9NXR1, amino acids 1–189) and Lis1 (Uniprot P43034, full-length). Predictions were performed using ColabFold v1.5.5, employing AlphaFold2 with MMseqs2, and conducted without a template. (**b**) The structure of IC2’s SAH and H2 (Uniprot Q13409, amino acids 1–67) predicated by ColabFold to interact with p150^Glued^’s CC1B (Uniprot Q14203, amino acids 357–589). (**c**) A proposed mechanism illustrating how IC functions as the central hub in dynein-1 activation, with further details described in the main text.

**Figure 4 cells-13-00330-f004:**
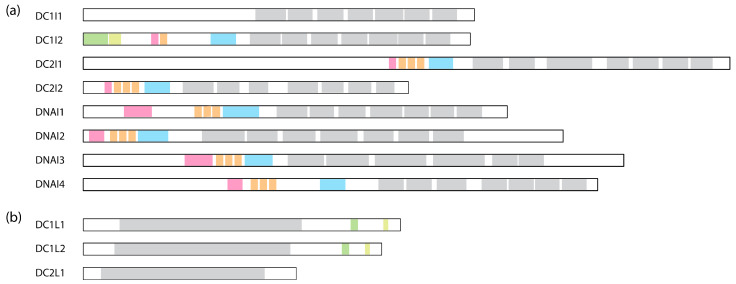
Comparison of human dynein IC and LIC subunits. (**a**) Comparison of the ICs containing the WD repeats. WD repeats are shown in grey, the Roadblock binding site is in blue, the LC8 binding site is in orange, the Tctex binding site is in pink, SAH is in green, and H2 is in light green. The binding sites are assigned based on cryo-EM structures: DC1I2 [6]: PDB 5NVU; DC2I1 and DC2I2 [8]: PDB 6SC2; DNAI1, DNAI2, DNAI3, and DNAI4 [10]: PDB 8J07. (**b**) Comparison of the LICs. The Ras-like domain is shown in grey; helix-1 is in green; and helix-2 is in light green. These regions are assigned based on AlphaFold structures deposited on Uniprot.

**Figure 5 cells-13-00330-f005:**
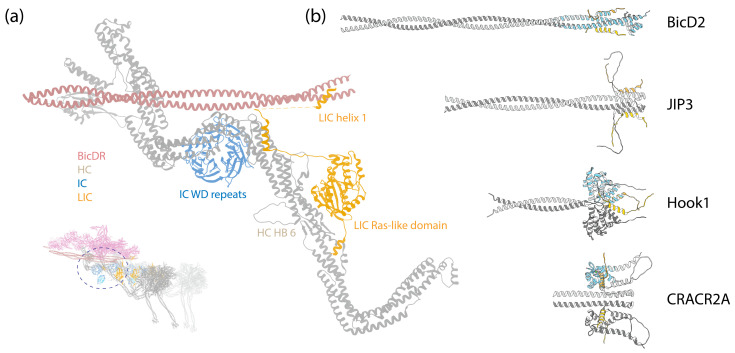
(**a**) A zoom-in view of the LIC binding to the HC. Inset: The DDR complex [67] (PDB 7Z8F); the circle indicates the zoom-in region. (**b**) ColabFold-predicted structures of LIC1 helix-1 interacting with various adaptors. These predictions were made in the same manner as those in Figure 3. The sequences analyzed include LIC1 helix-1 (Uniprot Q9Y6G9-1, amino acids 424–459); BicD2 (Uniprot Q8TD16-1, amino acids 1–240); JIP3 (Uniprot Q9UPT6-1, amino acids 1–180); Hook1 (Uniprot Q9UJC3-1, amino acids 1–239); CRACR2A (Uniprot Q9BSW2-2, amino acids 1–240). The adaptors are depicted in grey, and LIC1 helix-1 is in gold. The crystal structures of BicD2 (PDB 6PSE), Hook3 (PDB 6B9H), and CRACR2A (PDB 6PSD) with LIC1 helix-1 (blue) were structurally aligned to the predicted structures, using UCSF Chimera [112].

**Figure 6 cells-13-00330-f006:**
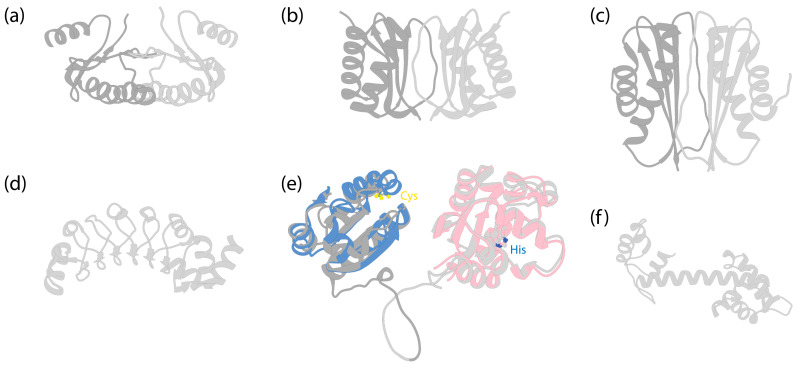
Structures of dynein LCs. (**a**) Solution structure of dimeric human Roadblock-1 [145] (PDB 1Z09). (**b**) Crystal structure of dimeric *Drosophila melanogaster* LC8 [146] (PDB 3BRI). (**c**) Crystal structure of dimeric *Drosophila* Tctex-1 [147] (PDB 1YGT). (**d**) Crystal structure of *Chlamydomonas* LC1 [148] (PDB 5YXM). (**e**) AlphaFold-predicted human TXND6 structure. Dark grey is for the thioredoxin domain, and light grey is for the nucleoside diphosphate kinase domain. The disordered N-terminal amino acids 1–10 and C-terminal amino acids 301–330 were removed for clarity. Blue indicates the predicted structure of human thioredoxin (Uniprot P10599), which was structurally aligned to TXND6, using UCSF Chimera [112]. LC3 and LC5 of *Chlamydomonas* [10] and LC3BL of *Tetrahymena* [9,45] have similar folds (not shown). The two cysteine residues in the -Cys-X-X-Cys- motif are highlighted by sulfur atoms (yellow) (Cys39 and Cys42 in TXND6; Cys32 and Cys35 in thioredoxin). Pink indicates the predicted structure of human nucleoside diphosphate kinase A (NDKA, Uniprot P15531), with amino acids 136–152 removed for clear depiction, and structurally aligned to TXND6. The conserved histidine residue is highlighted by the nitrogen atom (blue) (His279 in TXND6; His118 in NDKA). (**f**) AlphaFold-predicted structure of *Chlamydomonas* LC4 (Uniprot Q39584).

**Figure 7 cells-13-00330-f007:**
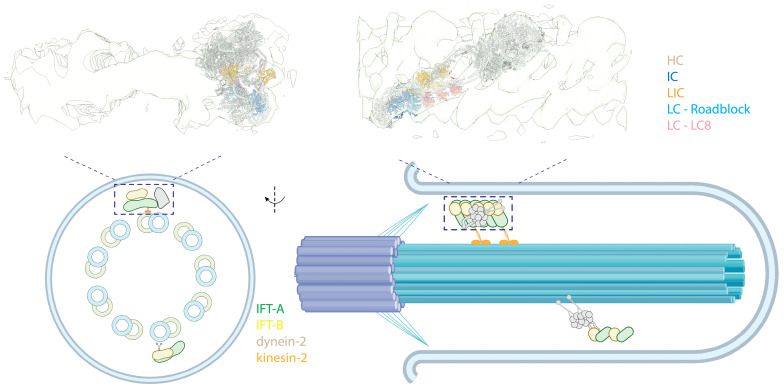
Intraflagellar transport. (**Top**) Side (**left**) and front (**right**) views of an autoinhibited human dynein-2 complex docked into a cryo-EM map of *Chlamydomonas* anterograde IFT complexes [8,204] (PDB 6SC2; EMD-4303). The colored text indicates the subunits. (**Bottom**, **left**) A cross-sectional illustrative view of a 9 + 0 immotile cilium. (**Bottom**, **right**) Side view of the cilium showing dynein-2 in an autoinhibited state, transported as cargo by kinesin-2 in anterograde trains.

**Figure 8 cells-13-00330-f008:**
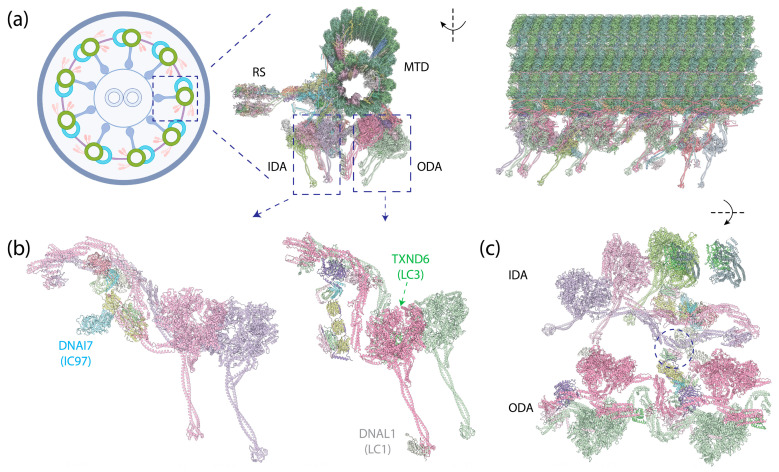
Human epithelial ciliary axoneme structure [10] (PDB 8J07). (**a**) (**Left**): An illustration of the cross-section of the motile axoneme. Middle: The cross-section view of the structure of an MTD with associated protein complexes. (**Right**): The side view of this structure. (**b**) (**Left**): IAD*f* structure, with an additional IC (DNAI7) on the side of the IC-LC tower. (**Right**): OAD structure, with DNAL1 (LC1) binding to the MTBD of DYH5 and TXND6 (LC3) binding to the neck region of DYH9. (**c**) Top view of the IDAs and ODAs. The circle indicates the interaction between OAD’s LC Tctex heterodimer and IAD*f*’s DNAI3 (IC140) and HC tail.

**Table 1 cells-13-00330-t001:** Composition of human dyneins’ non-catalytic subunits. The subunits of axonemal dyneins are based on the cryo-EM structure of human respiratory cilia [10] and follow the consensus nomenclature for dynein subunits [43]. It is important to note that the composition might vary depending on the cell types. For the composition of axonemal dyneins in *Chlamydomonas reinhardtii*, see ref. [44]. For the composition of axonemal outer-arm dyneins in *Tetrahymena thermophila*, see refs. [9,45].

Gene	Protein	Dynein Family	Uniprot Entry	Length (aa)
**IC**				
DYNC1I1	Cytoplasmic dynein 1 intermediate chain 1 (DC1I1, or IC1) ^1^	Dynein-1	O14576	645
DYNC1I2	Cytoplasmic dynein 1 intermediate chain 2 (DC1I2, or IC2)	Dynein-1	Q13409	638
DYNC2I1	Cytoplasmic dynein 2 intermediate chain 1 (DC2I1, or WDR60)	Dynein-2	Q8WVS4	1066
DYNC2I2	Cytoplasmic dynein 2 intermediate chain 2 (DC2I2, or WDR34)	Dynein-2	Q96EX3	536
DNAI1	Dynein axonemal intermediate chain 1 (DNAI1)	OAD	Q9UI46	699
DNAI2	Dynein axonemal intermediate chain 2 (DNAI2)	OAD	Q9GZS0	605
DNAI3	Dynein axonemal intermediate chain 3 (DNAI3, or WDR78, IC140)	IAD*f*	Q8IWG1	891
DNAI4	Dynein axonemal intermediate chain 4 (DNAI4, or WDR63, IC138)	IAD*f*	Q5VTH9	848
DNAI7	Dynein axonemal intermediate chain 7 (DNAI7, or Las1, IC97) ^2^	IAD*f*	Q6TDU7	716
**LIC**				
DYNC1LI1	Cytoplasmic dynein 1 light intermediate chain 1 (DC1L1, or LIC1)	Dynein-1	Q9Y6G9	523
DYNC1LI2	Cytoplasmic dynein 1 light intermediate chain 2 (DC1L2, or LIC2)	Dynein-1	O43237	492
DYNC2LI1	Cytoplasmic dynein 2 light intermediate chain 1 (DC2L1, or LIC3)	Dynein-2	Q8TCX1	351
**LC**				
DYNLT1	Dynein light chain Tctex-type 1 (DYLT1, or Tctex-1)	Dynein-1 Dynein-2 OAD & IAD*f*	P63172	113
DYNLT2B	Dynein light chain Tctex-type protein 2B (DYT2B, or Tctex1D2)	Dynein-2 OAD & IAD*f*	Q8WW35	142
DYNLT3	Dynein light chain Tctex-type 3 (DYLT3, or Tctex-3)	Dynein-1 Dynein-2	P51808	116
DYNLL1	Dynein light chain 1, cytoplasmic (DYL1, or LC8-1)	Dynein-1 Dynein-2 OAD & IAD*f*	P63167	89
DYNLL2	Dynein light chain 2, cytoplasmic (DYL2, or LC8-2)	Dynein-1 Dynein-2 OAD & IAD*f*	Q96FJ2	89
DNAL4	Dynein axonemal light chain 4 (DNAL4)	OAD	O96015	105
DYNLRB1	Dynein light chain roadblock-type 1 (DLRB1, or Roadblock-1)	Dynein-1 Dynein-2 OAD & IAD*f*	Q9NP97	96
DYNLRB2	Dynein light chain roadblock-type 2 (DLRB2, or Roadblock-2)	Dynein-1 Dynein-2 OAD & IAD*f*	Q8TF09	96
DNAL1	Dynein axonemal light chain 1 (DNAL1, or LC1)	OAD	Q4LDG9	190
NME9	Thioredoxin domain-containing protein 6 (TXND6, or LC3)	OAD	Q86XW9	330
DNALI1	Axonemal dynein light intermediate polypeptide 1 (IDLC, or p28)	IAD	O14645	258

^1^ The abbreviated name and alternative names commonly used are included in parentheses. ^2^ This IC does not have WD repeats and binds differently compared to other ICs.

## Data Availability

The ColabFold-predicted structures are available upon request.

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
