# Peer review of "Structure and Function of Dynein’s Non-Catalytic Subunits"

_cells, 2024, doi:10.3390/cells13040330_

Round 1
Reviewer 1 Report
Comments and Suggestions for Authors
This is a very timely review focused on the many non-catalytic subunits that comprise cytoplasmic and axonemal dyneins. In general, the manuscript is very comprehensive and well written. It provides a very nice description of our current understanding of these components. Given the comprehensive nature of this review and the highly conserved nature of the various dyneins, it is a little disappointing that the original identifications of some components in cytoplasmic dynein 1 (e.g. DYNLL1/2 aka LC8, and Tctex1), and cytoplasmic dynein 2 (e.g. WDR60 and WDR34) are not referenced.
l.16 conservation not conversation.
l.68 Table 1 is apparently missing the outer arm DNAL4 light chain that is homologous to DYNLL1/2 – see Braschi et al 2022 for most recent nomenclature of all known human dynein components.
l.452 LC8 was originally identified as an axonemal dynein subunit and subsequently found in mammalian cytoplasmic dynein. After that, it was found in many other systems. In contrast, Tctex1 was first identified in cytoplasmic dynein and subsequently found in axonemal dynein.
l.695 The motor unit of the outermost HC of three-headed outer arm dyneins is paralogous to the motor domain of the middle HC.
l.760 The structure of LC1, the most highly conserved dynein-specific component, was originally solved by NMR in 2000.
l. 773 It might be worth pointing out that alterations in redox state have been demonstrated to alter Chlamydomonas flagellar motility in vivo, and to control the sign of phototaxis which is driven by differential motility of the two flagella.
l.785 LC4 not only has EF hand motifs but has been demonstrated to directly bind Ca2+.
l.799 Dynein is an “ancient protein complex” not just a single protein.
Author Response
This is a very timely review focused on the many non-catalytic subunits that comprise cytoplasmic and axonemal dyneins. In general, the manuscript is very comprehensive and well written. It provides a very nice description of our current understanding of these components. Given the comprehensive nature of this review and the highly conserved nature of the various dyneins, it is a little disappointing that the original identifications of some components in cytoplasmic dynein 1 (e.g. DYNLL1/2 aka LC8, and Tctex1), and cytoplasmic dynein 2 (e.g. WDR60 and WDR34) are not referenced.
We appreciate the reviewer's insightful comments and their positive assessment of our review article. We acknowledge the oversight in not referencing the original identifications of certain components in cytoplasmic dynein-1 and cytoplasmic dynein-2. In response to this feedback, we have taken immediate steps to correct this omission. We have now incorporated references to properly attribute the original identifications of these subunits in our manuscript. We believe this addition enhances the completeness and accuracy of our review, and we are grateful to the reviewer for bringing this matter to our attention.
l.16 conservation not conversation.
Corrected at Line 17.
l.68 Table 1 is apparently missing the outer arm DNAL4 light chain that is homologous to DYNLL1/2 – see Braschi et al 2022 for most recent nomenclature of all known human dynein components.
We thank the reviewer for bringing this omission to our attention. We have incorporated the subunit into Table 1, with the reference added to the table caption (Lines 80–81). Furthermore, we have made necessary adjustments in the text, specifically within the section on the light chains of axonemal dynein (Lines 768-769) to ensure accuracy and completeness.
l.452 LC8 was originally identified as an axonemal dynein subunit and subsequently found in mammalian cytoplasmic dynein. After that, it was found in many other systems. In contrast, Tctex1 was first identified in cytoplasmic dynein and subsequently found in axonemal dynein.
We have now included references for the discovery of LC8 at Lines 464–465 and for Tctex at Lines 475–476, as well as references for the discovery of Roadblock at Lines 417–418, and WDR60 and WDR34 at Lines 602–603, in order to accurately reflect the historical context of these subunits. Thank you for pointing out these details.
l.695 The motor unit of the outermost HC of three-headed outer arm dyneins is paralogous to the motor domain of the middle HC.
Added at Line 717.
l.760 The structure of LC1, the most highly conserved dynein-specific component, was originally solved by NMR in 2000.
Added at Lines 781–782.
I.773 It might be worth pointing out that alterations in redox state have been demonstrated to alter Chlamydomonas flagellar motility in vivo, and to control the sign of phototaxis which is driven by differential motility of the two flagella.
Added at Lines 797–799.
l.785 LC4 not only has EF hand motifs but has been demonstrated to directly bind Ca2+.
Added at Lines 811–812.
l.799 Dynein is an “ancient protein complex” not just a single protein.
Corrected at Line 825.
Reviewer 2 Report
Comments and Suggestions for Authors
This is a comprehensive review that attempts to cover all of the aspects of cytoplasmic and axonemal dynein composition, structure, regulation and function. While this is certainly of value to the field, by undertaking this task on such a broad topic the authors make some serious inaccuracies (most from generalizing) and occasionally fail to refer to very relevant work in their statements.
-In the first sentence of the abstract: “…in eukaryotic cells”. This seems to be too general, taking into consideration that most plants do not have dyneins as you state in line 802.
-Mistake in the abstract, line 16: “…enabling it to perform critical tasks despite the conversation of its heavy chains”. I believe the authors mean “conservation”.
-In line 27, the authors state that …” axonemal dyneins, involved in the formation and sliding of axonemal doublet microtubules that drive flagellar beating”. Given that axonemal extension still occurs without these dyneins, the authors should rephrase to “…axonemal dyneins form the inner and outer rows of arms associated with the doublet microtubules of motile cilia”.
-In lines 28/29, the sentence is a bit misleading because not all dyneins are composed of HC dimers, i.e., some axonemal dyneins contain 3 HCs, as you state in the final sections of the review.
-In the Introduction, the authors go through the ATP hydrolysis cycle of dynein-1 and the function of each of its AAAs. Can they generalize? Has this been shown to be 100% conserved for the HCs of cytoplasmic dynein-2 and axonemal dyneins?
-On the first mention of dynein “adaptor” in line 88, the authors should add the word “cargo” just before (at least this first time here) to make it clearer to non-specialist readers.
- Is it possible to use a different color for LICs in Figure 1 a) and b)? The currently used makes it hard to clearly distinguish them from the HCs.
-“Nde1/Ndel1 (hereafter collectively referred to as Nde)” in lines 152/153: While NDE1/NDEL1 are paralogs with high similarity and some overlapping functions, there are numerous studies in the literature that show that they have specific functions and are differentially expressed in various tissues. Thus, it is widely misleading to authors to just refer to them as Nde! Furthermore, the literature used has to be carefully interpreted depending on what is actually shown rather than generalizing findings using one of the paralogs.
-Line 190: “Lis1,a dynein activator, enhances the assembly of DDA complexes”. Can you elaborate a bit further? I.e., as far as I understand, the literature shows that Lis1 stabilizes the open conformation of dynein-1, preventing it from naturally returning to the autoinhibited phi-particle state.
- References should be added to the sentence in lines 206-209
-Line 247 “As the only retrograde transport motor, dynein-1 performs a surprisingly wide range of functions.” Why use “only” here when right after you describe dynein-2, another retrograde transport motor… again the authors should be careful with this kind of misleading statements.
-Lines 328 - authors say “studies” but only cite one example.
-“Figure 4. Comparison of human dynein IC and LIC homologs”: The word “homologs” should be replaced by “subunits”. In the same legend, in (b), LIC should be corrected to “LICs”
-Line 360, “In mammals, there are two homologs of LIC” there are actually 3 LICs in mammals… again, the authors have to be careful in how they make their statements in such a comprehensive review. Simply include dynein-1 in the sentence and replace “homologs” by “subunits” or “forms”.
-The section starting in line 500: The authors should state somewhere that cytoplasmic dynein-2 has also been known as IFT dynein and dynein-1b (in Chlamydomonas). Also, it should be clear that the structure of IFT trains shown is not from human proteins.
-In line 510, add “IFT” before “trains”.
-In Figure 7 it is a bit confusing to show the BBS complex only in retrograde trains. As its structure/binding site when loaded into anterograde and retrograde trains remains unknown, I think it's best to just exclude it from the figure.
-In lines 546 and 547 the authors state “…mutations in the motor domain are sufficient to activate dynein-2.” They should rephrase to “… mutations in the linker domain are sufficient to activate dimers of dynein-2 motor domains in vitro”. At the end of the sentence, authors should also add the reference PMID: 28394326. Then the authors should also elaborate on recent work done with these dynein-2 mutations in vivo: PMID: 37883232.
-In line 574, “… trains turn around the tip of cilia…” should add “at” to make sense
-In lines 580/581: “Such a process could 580 release dynein-2 from the trains and activate dynein-2”. Should be rephrased to “Such a process could promote the remodeling of IFT trains and activate dynein-2 for IFT turnaround.”
-At the end of line 603, reference PMID: 31451806 should be added.
-Line 613, typo “Dynine-2”
-Line 631 to 635. It has been shown in numerous studies that LIC3 is essential for dynein-2 HC stability. The authors should state this and at least add the following reference PMID: 26077881, which clearly shows this function in human cells.
- Lines 725-730. Add reference(s).
Comments on the Quality of English LanguageJust a few typos already highlighted above.
Author Response
This is a comprehensive review that attempts to cover all of the aspects of cytoplasmic and axonemal dynein composition, structure, regulation and function. While this is certainly of value to the field, by undertaking this task on such a broad topic the authors make some serious inaccuracies (most from generalizing) and occasionally fail to refer to very relevant work in their statements.
We thank the reviewer for their valuable feedback. While our intention was to provide a comprehensive overview of cytoplasmic and axonemal dyneins, we recognize the importance of maintaining precision in our statements. We have carefully reviewed and modified the manuscript to address these comments, and have added references to enhance the accuracy and clarity of our review.
-In the first sentence of the abstract: “…in eukaryotic cells”. This seems to be too general, taking into consideration that most plants do not have dyneins as you state in line 802.
Corrected at Line 8.
-Mistake in the abstract, line 16: “…enabling it to perform critical tasks despite the conversation of its heavy chains”. I believe the authors mean “conservation”.
Corrected at Line 17.
-In line 27, the authors state that …” axonemal dyneins, involved in the formation and sliding of axonemal doublet microtubules that drive flagellar beating”. Given that axonemal extension still occurs without these dyneins, the authors should rephrase to “…axonemal dyneins form the inner and outer rows of arms associated with the doublet microtubules of motile cilia”.
We appreciate the reviewer’s suggestion. We have revised the text accordingly as recommended at Lines 28–29.
-In lines 28/29, the sentence is a bit misleading because not all dyneins are composed of HC dimers, i.e., some axonemal dyneins contain 3 HCs, as you state in the final sections of the review.
We appreciate the reviewer’s observation. To provide clarify on this matter, we have added a sentence at Lines 32–33 in the manuscript.
-In the Introduction, the authors go through the ATP hydrolysis cycle of dynein-1 and the function of each of its AAAs. Can they generalize? Has this been shown to be 100% conserved for the HCs of cytoplasmic dynein-2 and axonemal dyneins?
We are grateful to the reviewer for their insightful comments. In response, we have made the following adjustments to the manuscript:
- Cases that have only been studied in the context of dynein-1 are now specified accordingly.
- References have been updated to include studies across dynein families.
- We have clarified that the allosteric regulation of AAA1 by AAA3 and AAA4 is specific to dynein-1 (Lines 44–45).
- We have added a sentence to emphasize that fewer details are known for dynein-2 and axonemal dyneins (Lines 47–51).
- The statement regarding "linker docking on AAA5" has been modified to "linker docking on the ring domain" to account for the slight differences observed in axonemal dyneins (Line 54).
- We have specified that communication between strut/buttress to stalk is specific to cytoplasmic dyneins (Lines 62–63).
These changes aim to provide a more accurate and comprehensive representation of the topic. Thank you for your valuable feedback.
-On the first mention of dynein “adaptor” in line 88, the authors should add the word “cargo” just before (at least this first time here) to make it clearer to non-specialist readers.
We have included the word “cargo” before the first mention of dynein “adaptor” at Line 99 in the manuscript.
- Is it possible to use a different color for LICs in Figure 1 a) and b)? The currently used makes it hard to clearly distinguish them from the HCs.
We agree with the reviewer’s observation. To improve clarity, we have adjusted the legend color for "LIC" in Figure 1, Figure 5, and Figure 7 as suggested.
-“Nde1/Ndel1 (hereafter collectively referred to as Nde)” in lines 152/153: While NDE1/NDEL1 are paralogs with high similarity and some overlapping functions, there are numerous studies in the literature that show that they have specific functions and are differentially expressed in various tissues. Thus, it is widely misleading to authors to just refer to them as Nde! Furthermore, the literature used has to be carefully interpreted depending on what is actually shown rather than generalizing findings using one of the paralogs.
We appreciate the reviewer’s clarification on the distinction between NDE1 and NDEL1. To address this, we have made the necessary adjustments in the paragraphs referring to these paralogs, specifying the particular paralog in accordance with the relevant (Lines 162–177 and Lines 200–221). Thank you for pointing this out.
-Line 190: “Lis1,a dynein activator, enhances the assembly of DDA complexes”. Can you elaborate a bit further? I.e., as far as I understand, the literature shows that Lis1 stabilizes the open conformation of dynein-1, preventing it from naturally returning to the autoinhibited phi-particle state.
We appreciate the reviewer’s insight into the role of Lis1 in dynein activation. It is indeed documented that Lis1 primarily stabilizes the open conformation of dynein-1. However, it’s important to note that Lis1’s mechanism in activating dynein-1 may involve additional, distinct factors, as indicated by the incomplete rescue in observed with the open dynein-1 mutant in fungi lacking Lis1 (∆Lis1). The literature on Lis1 is extensive and at times contradictory, but the statement that "Lis1 enhances the assembly of DDA complexes" is well-established. As the primary focus of this review is not the detailed mechanism of Lis1 activation of Lis1, we believe that referencing a recent and comprehensive review of Lis1 in the following sentence (Line 203) will provide interested readers with a more in-depth understanding of this topic.
- References should be added to the sentence in lines 206-209
We have added references at Line 220.
-Line 247 “As the only retrograde transport motor, dynein-1 performs a surprisingly wide range of functions.” Why use “only” here when right after you describe dynein-2, another retrograde transport motor… again the authors should be careful with this kind of misleading statements.
We thank the reviewer for this comment and now write “the primary” instead of “only” at Line 258.
-Lines 328 - authors say “studies” but only cite one example.
We have revised the sentence accordingly to address this concern (Line 339).
-“Figure 4. Comparison of human dynein IC and LIC homologs”: The word “homologs” should be replaced by “subunits”. In the same legend, in (b), LIC should be corrected to “LICs”
We have made the necessary corrections in the legend for Figure 4 as suggested (Line 270). Thank you for pointing out these issues.
-Line 360, “In mammals, there are two homologs of LIC” there are actually 3 LICs in mammals… again, the authors have to be careful in how they make their statements in such a comprehensive review. Simply include dynein-1 in the sentence and replace “homologs” by “subunits” or “forms”.
We have addressed this issue by adding "dynein-1" to the sentence (Line 371) for accuracy. However, we would prefer to retain the word "homologs" to avoid potential confusion, as each dynein HC indeed has two LICs and using “subunits” or “forms” might not fully convey this distinction. Thank you for your understanding.
-The section starting in line 500: The authors should state somewhere that cytoplasmic dynein-2 has also been known as IFT dynein and dynein-1b (in Chlamydomonas). Also, it should be clear that the structure of IFT trains shown is not from human proteins.
We have included the additional information that cytoplasmic dynein-2 is also known as IFT dynein and dynein-1b (in Chlamydomonas) at Line 530. Furthermore, we have modified the Figure 7 caption to clarify that the structure of IFT trains shown is not from human proteins (Line 541).
-In line 510, add “IFT” before “trains”.
We have added “IFT” before “Trains” as suggested (Line 525).
-In Figure 7 it is a bit confusing to show the BBS complex only in retrograde trains. As its structure/binding site when loaded into anterograde and retrograde trains remains unknown, I think it's best to just exclude it from the figure.
We appreciate the reviewer’s input regarding the inclusion of the BBsome in Figure 7. Given the uncertainty surrounding its structure and binding sites in both anterograde and retrograde trains, we concur that it is best to exclude it from the figure.
-In lines 546 and 547 the authors state “…mutations in the motor domain are sufficient to activate dynein-2.” They should rephrase to “… mutations in the linker domain are sufficient to activate dimers of dynein-2 motor domains in vitro”. At the end of the sentence, authors should also add the reference PMID: 28394326. Then the authors should also elaborate on recent work done with these dynein-2 mutations in vivo: PMID: 37883232.
We appreciate the reviewer’s suggestions. We have revised the sentences to specify that mutations in the linker domain activate dimers of dynein-2 motor domains in vitro (Lines 561–562). Accordingly, we have included the reference PMID: 28394326. We also include information on recent in vivo work with these dynein-2 mutations, referencing PMID: 37883232 (Lines 563–565). Thank you for bringing these details to our attention.
-In line 574, “… trains turn around the tip of cilia…” should add “at” to make sense
We have corrected this sentence (Line 592). Thanks!
-In lines 580/581: “Such a process could 580 release dynein-2 from the trains and activate dynein-2”. Should be rephrased to “Such a process could promote the remodeling of IFT trains and activate dynein-2 for IFT turnaround.”
We have made the requested rephrasing. Considering the mention of "remodeling of IFT trains" in the preceding sentence (Line 598), we have adjusted the sentence to read, "Such a process could activate dynein-2 for IFT turnaround" (Line 599). Thank you for the clarification.
-At the end of line 603, reference PMID: 31451806 should be added.
We have added the reference (Line 622).
-Line 613, typo “Dynine-2”
Thank you for pointing out this typo! (Line 632)
-Line 631 to 635. It has been shown in numerous studies that LIC3 is essential for dynein-2 HC stability. The authors should state this and at least add the following reference PMID: 26077881, which clearly shows this function in human cells.
We have included the information about LIC3's essential role in dynein-2 HC stability and referenced the study with PMID: 26077881 (Line 654). Thank you for highlighting this point.
- Lines 725-730. Add reference(s).
We have added references (Lines 747–753). Thank you for bringing this to our attention.
Round 2
Reviewer 2 Report
Comments and Suggestions for Authors
The authors have adequately addressed all of my concerns. This review is know ready to be published in Cells. Congratulations to the authors.